# *Hippeastrum stapfianum* (Kraenzl.) R.S.Oliveira & Dutilh (Amaryllidaceae) Ethanol Extract Activity on Acetylcholinesterase and PPAR-α/γ Receptors

**DOI:** 10.3390/plants11223179

**Published:** 2022-11-21

**Authors:** Kicia Karinne Pereira Gomes-Copeland, Cinthia Gabriel Meireles, João Victor Dutra Gomes, Amanda Gomes Torres, Simone Batista Pires Sinoti, Yris Maria Fonseca-Bazzo, Pérola de Oliveira Magalhães, Christopher William Fagg, Luiz Alberto Simeoni, Dâmaris Silveira

**Affiliations:** 1Laboratory of Natural Products, Faculty of Health Sciences, University of Brasilia, Brasília 70910-900, DF, Brazil; 2Laboratory of Molecular Pharmacology, Health Sciences Faculty, University of Brasilia, Brasília 70910-900, DF, Brazil; 3Faculty of Ceilandia, University of Brasilia, Brasília 70919-970, DF, Brazil

**Keywords:** acetylcholinesterase, antioxidant activity, nuclear receptors, alkaloids, Cerrado

## Abstract

*Hippeastrum stapfianum* (Kraenzl.) R.S.Oliveira & Dutilh (Amaryllidaceae) is an endemic plant species from the Brazilian savannah with biological and pharmacological potential. This study evaluated the effects of ethanol extract from *H. stapfianum* leaves on acetylcholinesterase enzyme activity and the action on nuclear receptors PPAR-α and PPAR-γ. A gene reporter assay was performed to assess the PPAR agonist or antagonist activity with a non-toxic dose of *H. stapfianum* ethanol extract. The antioxidant capacity was investigated using DPPH^•^ scavenging and fosfomolybdenium reduction assays. The identification of *H. stapfianum*‘s chemical composition was performed by gas chromatography–mass spectrometry (GC-MS) and HPLC. The ethanol extract of *H. stapfianum* activated PPAR-α and PPAR-γ selectively, inhibited the acetylcholinesterase enzyme, and presented antioxidant activity in an in vitro assay. The major compounds identified were lycorine, 7-demethoxy-9-*O*-methylhostasine, and rutin. Therefore, *H. stapfianum* is a potential source of drugs for Alzheimer’s disease due to its ability to activate PPAR receptors, acetylcholinesterase inhibition activity, and antioxidant attributes.

## 1. Introduction

Amaryllidaceae’s chemical constitution comprises a class of alkaloids characteristic of the family. Several studies exploring these compounds have drawn attention to them due to their different biological activities, such as anticancer and cytotoxic effects and acetylcholinesterase modulation [1]. Within this family, *Hippeastrum stapfianum* (Kraenzl.) R.S.Oliveira & Dutilh [*Hippeastrum goianum* (Ravena) Meerow] (Figure 1) stands out as a Brazilian endemic species once considered at risk of extinction [2]. A previous study by our group suggested that in vitro germination and micropropagation of *H. stapfianum* can improve lycorine biosynthesis, signifying that the obtained products had a more noticeable potential acetylcholinesterase inhibition [3].

Acetylcholinesterase (AChE) belongs to the cholinesterase family and is involved in the hydrolysis of the neurotransmitter acetylcholine (Ach) into choline and acetic acid, allowing a cholinergic neuron to return to the resting state after its activation [4]. AChE is found in many tissues such as central and peripheral tissues, motor and sensory fibers, and red blood cell membranes [5,6]. AChE’s primary role involves neuronal transmission and signaling between synapses, preventing ACh dispersal and activation of receptors and avoiding the amplification and propagation of cellular signaling [6]. In addition, AChE can interfere with the inflammation response, oxidative stress, apoptosis, and aggregation of pathological proteins [7].

Thus, inhibiting AChE can be useful, therapeutically speaking, for several health injuries, such as symptomatic treatment of glaucoma and myasthenia gravis [8,9], as well as the palliative treatment of neurodegenerative diseases, such as some types of sclerosis, depressive disorders, Parkinson’s, Huntington’s, and Alzheimer’s diseases [7,10].

A comprehensive molecular characterization of natural compound candidates showed the potential of some molecules in Alzheimer’s disease (AD) treatment due to acetylcholinesterase inhibition [11,12]. Galantamine is an example of this rational drug discovery race concerning Amaryllidaceae alkaloids and is already an approved drug for treating AD patients. Galantamine inhibits acetylcholinesterase through the allosteric modulation of nicotinic receptors, improving cognitive function and enhancing the signs of memory [13,14]. It is important to highlight that Alzheimer’s disease has been the foremost cause of dementia in recent years. According to the World Health Organization (WHO), dementia, including AD, is the seventh highest cause of death among all diseases worldwide [15]. In 2018, the prevalence of AD was about 50 million people, and the survival after diagnosis is about four years [16].

A review explored the mechanism of the superfamilies of peroxisome proliferator-activated receptors (PPAR) in AD, and despite the limited exploration of the molecular aspects, the findings showed anti-inflammatory properties mediated by PPAR activation [17]. PPARs are transcription factors of the nuclear receptor superfamily, activated by ligands. Specifically, the PPAR-γ subtype is expressed in the microglia and the astrocytes and has important roles in regulating inflammation and different pathways in the central nervous system [18]. In addition, the agonism of PPAR-γ is related to decreasing the tau aggregation and minimizing neuroinflammation [19]. PPAR-α is distributed in different hippocampus regions and is involved in the metabolism of amyloid-beta precursor protein in the brain, influencing, directly or indirectly, tau phosphorylation [20]. Thus, the positive modulation of PPAR-α and PPAR-ɣ might be important for improving brain cell metabolism and cognitive function [18,20]. 

A further specificity of the receptor superfamily is the capacity to regulate oxidative stress, energy homeostasis, and mitochondrial fatty acid metabolism [21,22]. The antioxidant and anti-inflammatory action, mediated by the PPAR superfamily role, have been described in several diseases such as diabetes type 2 [23] and traumatic and dental injuries [24]. All effects modify the neurodegenerative process by regulating neurotransmission and the interaction with amyloid-beta peptide deposition [25].

Evidence supports the fact that myelin damage/demyelination can influence AD pathogenesis, probably occurring in the early stages of the disease preceding the onset of typical neurological changes such as Aβ plaques and neurofibrillary tangles [26]. Thyroid hormone nuclear receptor beta (TR-β) is expressed in oligodendrocytes (OLs) and oligodendrocyte precursor cells (OPCs). OPCs are self-renewing cells that play an important role in the remyelination following demyelination [27].

Considering all these aspects, this study evaluated the effects of ethanol extract from *H. stapfianum* leaves on acetylcholinesterase activity and PPAR-α and PPAR-γ activation.

## 2. Results

### 2.1. EE Upregulates the PPAR-α and PPAR-γ Receptor

In order to verify the maximum non-toxic concentration of ethanol extract from *Hippeastrum stapfianum* leaves (EE), we evaluated the HeLa cells’ viability with an increasing concentration set of EE (200–1000 µg/mL) using an MTT assay, and we found an appropriate concentration under 300 µg/mL (around 90% cell viability; Figure 2A). This concentration was selected as the maximum concentration in the luciferase assay. The HeLa cells were co-transfected with plasmids containing the cDNA of GAL4-PPAR-α or GAL4-PPAR-γ or GAL4-TR to examine the EE ability to induce the PPAR-α, PPAR-γ, or TR receptor. The dose–response analysis suggested that the EE statistically improved the transcriptional activity of PPAR-α and PPAR-γ at 250 and 300 (μg/mL) compared to the vehicle (Figure 2B,C). However, compared with the positive control (rosiglitazone for PPAR-γ or bezafibrate for PPAR-α), the agonist activity was slight, suggesting partial modulation in response to these receptors. On the other hand, no activation of the TR-β receptor was detected with any of the concentrations tested of the EE (Figure 2D), suggesting a selective effect on PPAR receptors.

### 2.2. EE Performs Antioxidant Capacity and Promotes Acetylcholinesterase Inhibition In Vitro

The EE inhibited the AChE activity (IC_50_ = 27.07 ± 2.82 µg/mL). The experiment was controlled using galantamine hydrobromide as a positive inhibitor (IC_50_ = 0.32 ± 0.08 µg/mL). Although higher than the obtained for galantamine hydrobromide, the IC_50_ result for EE is relevant once an extract was used and not an isolated compound. The EE also showed antioxidant capacity through DPPH scavenging activity (EE EC_50_ = 61.88 ± 0.43 µg/mL; AA EC_50_ = 7.71 ± 0.45 µg/mL) and the reduction of the fosfomolybdenium complex (93.76 ± 6.58 µg of equivalents of AA per 1 mg of extract or fraction).

### 2.3. EE and Fractions Chemical Characterization by GC-MS Analysis and HPLC

The extract was fractionated for better chemical characterization. The EE and the three fractions (hexane, ethyl acetate, and ethyl acetate/methanol fractions) were analyzed by GC-MS. The fractionation increased the concentration of alkaloids in the ethyl acetate/methanol fraction (EAMF), and although alkaloids were not found in the extract, they were found in the fraction with the highest polarity; Amaryllidaceae alkaloids were detected only in the EAMF: lycorine and 7-demethoxy-9-*O*-methylhostasine, representing 0.81 and 94.55 %, respectively. Obtained data of three additional alkaloids (NI-1, NI-2, and NI-3) were compared with the literature data, but complete identification was not possible. The mass fragmentation, retention time, and % of total ion current (TIC) are described in Table 1, and the chromatogram is illustrated in Figure 3.

HPLC analysis of EE was performed to identify phenolic compounds in the extract in comparison to a library of flavonoids and other phenolics. Rutin was identified, as shown in Figure 4. 

In summary, EE positively modulates the PPAR receptors, inhibits the acetylcholinesterase activity, and shows antioxidant capacity (Figure 5) (Servier Medical Art by Servier is licensed under a Creative Commons Attribution 3.0 Unported License). The mechanisms might be related to lycorine, 7-demethoxy-9-*O*-methylhostasine, and rutin. 

## 3. Discussion

Alzheimer’s disease is characterized by the accumulation of two proteins, amyloid β (Aβ) and tau [31]. To extend Aβ deposits on the surface of the neuropils, the area where compacted dendrites, glial cells, and axon branches are found expands until the extracellular medium. Tau is already associated with the formation of a microtubule tangle that also involves the dendrites, preventing them from binding to the microtubules. They all contribute to forming these proteins’ plaques and tangles, which promote hyperphosphorylation and neuroinflammation and inhibit neurons from making normal synapses. These points are recognized as a major component AD’s physiopathology, contributing to neurodegeneration and, subsequently, the disease’s progression [32].

Pharmacological treatment represents a great challenge due to the complex physiopathology involved in AD. The therapy generally includes cognitive enhancement therapies, improving neuropsychiatric symptoms, and disease-modifying therapies for AD [31]. In drug discovery, multi-specific targets have been considered to be an important step in pre-clinical evaluations. In this sense, the single-target substitution for multi-target drugs in current medicinal chemistry defines good drug discovery practices [33]. Thus, natural products represent a strategic resource for the identification of molecules that specifically act on different pathological targets related to AD.

There is a hypothesis that PPARs are also involved in the pathogenesis of several central nervous system disorders, including AD [34]. Previous studies showed that agonist PPAR treatment led to the phenotypic polarization of microglial cells from a pro-inflammatory state to an anti-inflammatory state associated with enhanced phagocytosis of deposited forms of amyloid. The reduction in amyloid levels has been associated with a reversal of contextual memory deficits in drug-treated mice, and these data can explain how PPAR activation facilitates amyloid clearance and supports the therapeutic utility of PPAR agonists in the treatment of AD [32]. In the present study, we evaluate the EE’s effect on acetylcholinesterase inhibition and PPAR-α and PPAR-γ activation, which seems to occur selectively. In addition, EE was tested for antioxidant activity by two different models.

Preliminary results found in a study by Reich et al. (2018) demonstrated neuroprotection based on PPAR-δ plus PPAR-γ agonist treatments [35]. They found it ineffective in restoring choline acetyltransferase (ChAT) and fully normalizing myelin-associated glycoprotein (MAG-1) levels and AβPP-Aβ. In contrast to the findings of the present work, they showed increasing AChE in brain cultures.

Our findings support the literature, which observed the effect of the fractions from *Hippeastrum puniceum* (Lam.) Urb. and *Hippeastrum barbatum* Herb. in AChE inhibition, with IC_50_ values of 25.73 ± 1.75 g/mL and 28.13 ± 1.68 µg/mL, respectively. EE, also obtained from another *Hippeastrum* species, achieved similar results for AChE inhibition [36]. The results of AChE inhibition in micropropagated *H. stapfianum* without and with growth regulators showed IC_50_ values of 386.00 ± 0.97 µg/mL and 114.80 ± 0.95 µg/mL, respectively [3]. In addition, Gasca et al. (2020) showed that *Hippeastrum psittacinum* Herb. ethanol extract inhibited the AChE (IC_50_ = 11.2 μg/mL) better than the alkaloid levels, suggesting a potentiating effect of the compounds in the extract [37]. Moreover, Castillo et al. (2018) suggested that the AChE inhibitory activity of the EE of *Caliphruria subedentata* Baker bulbs (IC_50_ ~45 μg/mL) was due to the interaction among different alkaloids, such as lycorine, homolycorine, galantamine, and montanine [38].

In the present study, we highlight another finding: the EE antioxidant potential, evaluated by two in vitro models, DPPH scavenging with EC_50_ of 61.88 ± 0.43 µg/mL and phosphomolybdenum reduction (93.76 ± 6.58 µg of equivalents of AA per 1 mg of EE). Although this result seems irrelevant, it is important to point out that EE refers to a complex mixture of diverse compounds being compared to a pure substance (ascorbic acid). The antioxidant response corroborates the findings of Reich et al. (2018) that demonstrate the PPAR agonist treatment reduced the marker of oxidative stress by 8-hydroxydeoxyguanosine levels in brain cells relative to the control [35]. *Hippeastrum stapfianum* can biosynthesize lycorine, 7-demethoxy-9-*O*-methylhostasine, and rutin, identified in EE, and this result corroborates another study on the same species [29]. The data obtained here suggest that compounds from EE present a potential, although slight, PPAR activity. In addition, the results suggest acetylcholinesterase and oxidative stress modulation promoted by *Hippeastrum stapfianum* extract. 

Finally, the limitations of our experimental approach must be pointed out. Due to the limited availability of plant samples, performing the gene reporter assays with the fractions and isolated chemical compounds was not possible. The results found in the present study are preliminary, and there is a need to clarify which compounds in the extract contribute to acetylcholinesterase inhibition, oxidative stress modulation, and PPAR action.

*Hippeastrum stapfianum* ethanol extract (EE) can potentially lead to a new option for Alzheimer’s disease treatment due to the ability to activate PPAR receptors selectively, inhibit AChE, and present antioxidant capacity under the conditions evaluated. Therefore, these data suggest that *H. stapfianum* represents a promising therapeutic approach for treating AD, although further studies are needed to characterize the mechanism.

## 4. Materials and Methods

### 4.1. Hippeastrum stapfianum Plant Material

Adult *H. stapfianum* specimens were collected at Estrutural, Brasilia, DF, Brazil. The species was identified by Prof. Dr. Christopher William Fagg (University of Brasilia), and a dried sample was deposited in the Herbarium of the University of Brasília (UB) (voucher number UB 217068). This work was carried out under register 163E599 in the National Genetic Heritage Management System (SisGen, Brasilia, Brazil).

### 4.2. Hippeastrum stapfianum Extracts and Fractions

Leaves of *H. stapfianum* were dried in an oven with air circulation at 37 °C for 48 h. The plant material was homogenized and macerated in hexane P.A. (Vetec, Recife, Brazil) for 24 h. After maceration with hexane, a second extraction was carried out with ethanol (P.A., Vetec, Recife, Brazil) for 72 h, and the extracted solution was filtered and concentrated at 40 °C using a rotary evaporator under vacuum (Hei-VAP Advantage, ML, G1, 115v—Heidolph, Schwabach, Germany). The ethanol extract (EE) was stored in a freezer at −20 °C to be used in the subsequent bioassays and chemical fractionation. The ethanol extract was fractionated as previously described [36] to obtain alkaloid-rich fractions. The obtained fractions were hexane, ethyl acetate, and ethyl acetate/methanol fractions (HF, EAF, and EAMF, respectively), stored in a freezer at −20 °C to further analyzed by GC-MS.

### 4.3. HeLa Cell Culture

Human cervical cancer HeLa cells (BCRJ 0100; Rio de Janeiro Cell Bank, Rio de Janeiro, Brazil) were grown in DMEM medium (Dulbecco’s modified Eagle medium—Sigma-Aldrich, Jurubatuba, Brazil) supplemented with glutamine (2 nM; Sigma-Aldrich), penicillin–streptomycin (100 IU/mL; 100 µg/mL; Sigma-Aldrich), and 10% (*v*/*v*) fetal bovine serum (FBS, Gibco, Thermo Fisher Scientific, Waltham, MA, USA), at 37 °C in a 5% CO_2_ humidified atmosphere.

### 4.4. Cell Viability by MTT Assay (Mitochondrial Activity)

Human HeLa cells (5 × 10^4^/well; 70~90% confluence) were seeded into 96-well plates and incubated in a standard medium overnight before treatment of increasing concentrations (200 to 1000 µg/mL) of EE for 24 h. An MTT solution (1 mg/mL, Sigma-Aldrich) was added, and the plate was then incubated at 37 °C for 4 h before absorbance measurement at 590 nm, using a Multimode Plate Reader (EnSpire, Perkin-Elmer, Singapore). Cell viability was calculated as a percentage of the untreated control cells.

### 4.5. Transfection and Luciferase Gene Reporter Assay

Human HeLa cells (2.5 × 10^4^/well, in 48 wells) were co-transfected with expression vectors containing cDNA (60 ng) for chimeric nuclear receptors, including the ligand binding domain of human (LBD) of PPAR-y of GAL4 yeast transcription factor and a responsive element GAL4 fused to luciferase reporter gene, or PPAR-α or TR-β fused to the DNA-binding domain (DBD). Transfections were performed with Lipofectamine (Lipofectamine 2000 Transfection Reagent, Thermo Fischer Scientific) according to the manufacturer’s instructions. Cells were exposed for 24 h to the vehicle (negative control (DMSO 0.1% *v*/*v*, Sigma-Aldrich); rosiglitazone 10 μM (Cayman Chemical, Ann Harbor, USA), positive control, an agonist of PPAR-y; bezafibrate 100 μM (Sigma-Aldrich), positive control, an agonist of PPAR-α; thyroid hormone (T3, Sigma-Aldrich) 10 μM, positive control, agonist activity in TR-β; and EE (100 to 300 µg/mL). Luciferase activity was measured in a luminometer (GloMax^®^ 20/20 Luminometer—Promega, São Paulo, Brazil) with a luciferase reporter assay kit (Promega). Results were reported as luciferase activity induced by the samples and controls relative to the vehicle (DMSO). The experiment was performed in triplicate.

### 4.6. Acetylcholinesterase Inhibition

The acetylcholinesterase inhibition assay was performed based on the modified methodology of Ellman et al. (1961) [39,40]. Briefly, 50 μL of 0.25 U/mL enzyme in phosphate buffer (PBS) (8 mM K_2_HPO_4_ (Vetec, Recife, Brazil), 2.3 mM NaH_2_PO_4_ (Synth, Diadema, Brazil)_,_ and 0.15 M NaCl (Vetec, Recife, Brazil), pH 7.5), 50 μL of PBS buffer, and 50 μL of EE in different concentrations (solubilized in 15% methanol, Vetec, Recife, Brazil) were added to the 96-well microplates. This mixture was incubated for 30 min at 37 °C. Then, 100 μL of substrate were solubilized in distilled water q.s.p. 20 mL (0.24 mM acetylthiocholine iodide 0.04 M, Na_2_HPO_4_ and 0.2 mM DTNB, all purchased from Sigma-Aldrich, Jurubatuba, Brazil) and added to the 96-well plate. The mixture was again incubated for 10 min at 37 °C. Subsequently, the absorbance reading of each triplicate was performed at 405 nm using a Multimode Plate Reader (EnSpire, Perkin-Elmer, Singapore). Inhibitory activity was calculated as a percentage. Galantamine hydrobromide was used as a positive control (0.01–2.0 μg/mL).

### 4.7. 2,2-Diphenyl-1-picrylhydrazyl (DPPH) Scavenging Assay

The antioxidant capacity was investigated by a DPPH^•^ radical scavenging in vitro assay [41]. The activity of the standard or EE was determined by adding 100 μL of ethanol 95% (*v*/*v*) (Vetec, Recife, Brazil), 100 μL of sodium acetate buffer 100 mM (pH 5.5, Merck,, Darmstadt, Germany), 50 μL of 507.2 μM DPPH (Sigma-Aldrich, Jurubatuba, Brazil) radical solution in ethanol, and 10 μL of standard or sample to 96-well plate. After 15 min, the spectrophotometer was read, discounting the blank. For the blank, 150 μL of ethanol 95% (*v*/*v*), 100 μL of sodium acetate buffer (pH 5.5), and 10 μL of standard or sample were added in the same concentration. The absorbance was determined at 517 nm using a Multimode Plate Reader (EnSpire, Perkin-Elmer, Singapore). Ascorbic acid (AA, Merck, Darmstadt, Germany) was used as positive control (0.6–6 µg/mL). The results were expressed as the efficient concentration that can scavenge DPPH^•^ radical at 50% (EC_50_). All analyses were performed in triplicate, and data were expressed as mean ± standard deviation.

### 4.8. Fosfomolybdenium Reduction Assay

EE was evaluated for the reductive capacity of fosfomolybdenium [42], with some modifications [43]. The reagent solution was freshly prepared by mixing 11.2 mL of 28 nM anhydrous monobasic sodium phosphate (Sigma-Aldrich), 10 mL of 0.6 M sulfuric acid (Dinamica, Indaiatuba, Brazil), 6.4 mL of 4 nM ammonium molybdate (Merck), and 12.4 mL of distilled water. Then, 1.0 mL of reagent solution and 0.1 mL of sample (EE or AA) were added to individual microtubes and heated at 95 °C for 90 min in a water bath. Next, 250 µL from each microtube was transferred to a microplate well, and the absorbance was read at 695 nm using a Multimode Plate Reader (EnSpire, Perkin-Elmer, Singapore). Samples were evaluated in triplicate. The activity was expressed as µg of equivalents of AA per 1 mg of extract or fraction.

### 4.9. GC-MS Analysis

A total of 4 mg of EE, HF, EAF, and EAMF was dissolved in 1 mL of MeOH (Tedia, Rio de Janeiro, Brazil) and/or CHCl_3_ (Tedia, Rio de Janeiro, Brazil) and injected directly into the GC-MS apparatus (Clarus 680 GC, Perkin Elmer) coupled to a quadrupole mass spectrometer (Clarus SQ8 MS, Perkin Elmer, Singapore). Perkin Elmer Elite-5MS capillary column (length 30 m × inner diameter 0.25 mm × film thickness 0.25 µm) was used. The temperature gradient was performed as follows: 12 min at 100 °C, 100–180 °C at 15 °C/min, 180 at 300 °C at 5 °C/min, and 10 min at 300 °C. The injector and detector temperatures were 280 and 250 °C, respectively, and the carrier gas flow rate (He) was 1 mL/min. A 1:5 split ratio was applied, and the injection volume was 1 μL. Alkaloids were identified by comparing their mass spectra and retention index (RI). Mass spectra were analyzed using AMDIS 2.64 software (NIST) (Gaithersburg, USA), and RI was recorded with a calibration mixture of hydrocarbon standards (C_9_–C_36_). The proportion of each alkaloid present in extracts and fractions analyzed by GC-MS was expressed as a percentage of the alkaloid peak area as a function of the total ion current (TIC).

### 4.10. HPLC-DAD-UV Analysis

EE was solubilized in methanol (Tedia) (1 mg/mL), and 10 μL of this sample was analyzed using LaChrom Elite HPLC system (Hitachi, Tokyo, Japan) liquid chromatograph equipped with L2130 pump, auto-sampler L2200, L2300 column oven was set at 25 °C and an L2455 diode array detector (DAD) (Hitachi, Tokyo, Japan). The C-18 column (5 μm, 150 mm × 4.6 mm) was used in combination with an appropriate guard column (4.0 mm × 4.0 mm; 5 μm of particle size) (Merck, Darmstadt, Germany). The analysis was performed at a wavelength fixed at 354 nm. The eluents used were aqueous phosphoric acid (Merck, Darmstadt, Germany) (1%) (solvent A) and acetonitrile (J. T. Baker, Leicestershire, UK) (solvent B). The gradient employed was 90% A and 10% B for 0 min, 70% A and 30% B for 40 min, 50% A and 50% B for 50 min, 90% A and 10% B for 51 min, and 90% A and 10% B for 55 min at a flow rate of 0.6 mL/min. Data acquisition was performed using ExChrom Elite software (version 3.3.2 SP1) (Scientific Software Inc., Santa Clara, CA, USA). The compounds present in the extract were compared according to their UV–Vis spectra (similarity index > 0.99) and retention times with commercial standards, as previously described [44].

### 4.11. Statistical and Data Analysis

The viability of the cells was assessed by mitochondrial activity presented by percentage related to DMEM media (control). The gene reporter luciferase assay results were presented in a dot plot with standard deviation (SD). Because some groups presented non-normally distributed data, statistical differences among groups were tested by Kruskal–Wallis and Dunn’s post hoc test (GraphPad Prism Software, version 5.01). The activation rate of the transcription of the groups treated with the positive controls or the extract was compared to the group treated with vehicle (control). The significance criterion for all analyses was *p* < 0.05. The in vitro acetylcholinesterase inhibition and antioxidant capacity assays were tested by one-way analysis of variance (ANOVA) followed by Dunnett’s post hoc test, and *p* values < 0.05 were considered significant.

## Figures and Tables

**Figure 1 plants-11-03179-f001:**
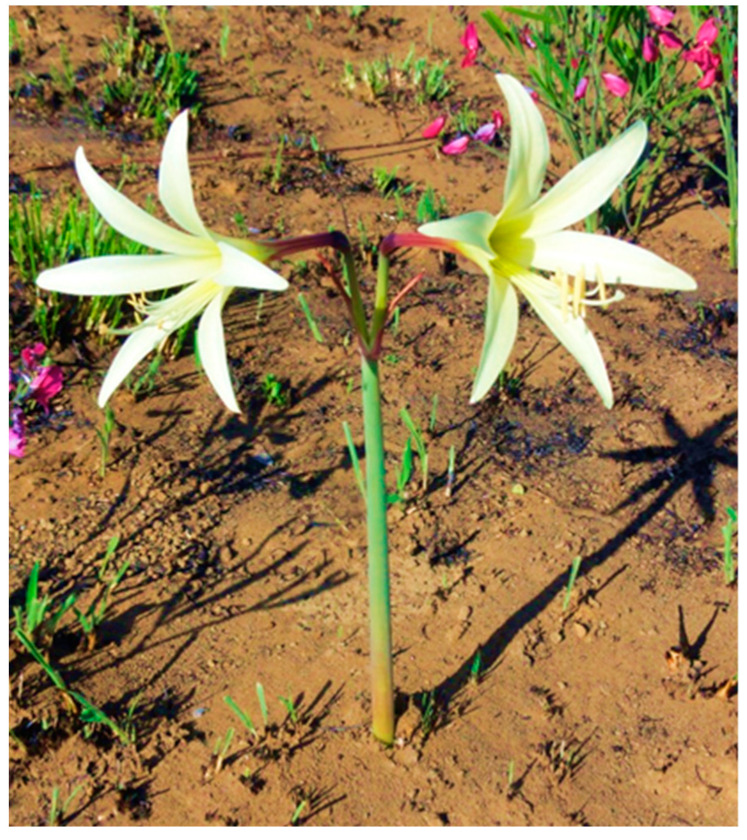
*Hippeastrum stapfianum* flowers in Brazilian savannah (Cerrado).

**Figure 2 plants-11-03179-f002:**
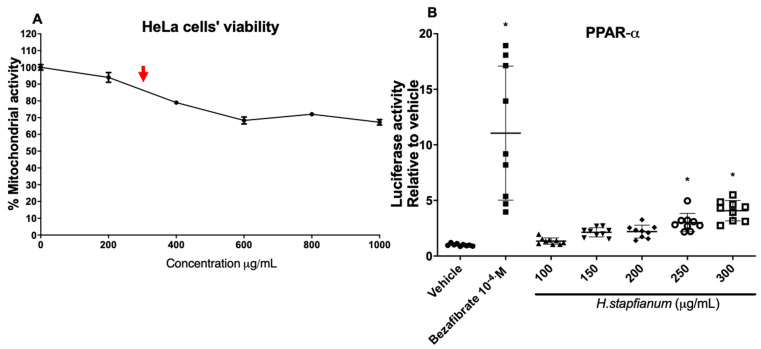
HeLa cells’ viability under treatment with ethanol extract of *H. stapfianum* Ravena (Amaryllidaceae) leaves (EE), and the action of the same EE on nuclear receptors PPAR\α and PPAR-γ. (**A**) HeLa cells were treated with different concentrations of EE (200–1000 µg/mL) for 24 h. Cell viability was measured with MTT assay and appropriate concentration under 300 µg/mL (red arrow) (**B**) Increasing concentrations of EE (100–300 µg/mL), vehicle (DMSO 0.1% *v*/*v*), or positive control (bezafibrate 100 μM) on PPAR-α transcriptional activity in HeLa cells, measured with luciferase gene reporter assay. (**C**) Increasing concentrations of EE (100–300 µg/mL), vehicle (DMSO 0.1% *v*/*v*), or positive control (rosiglitazone 10 μM) on PPAR-γ transcriptional activity in HeLa cells, measured with luciferase gene reporter assay. (**D**) Increasing concentrations of EE (100–300 µg/mL), vehicle (DMSO 0.1% *v*/*v*), or positive control PC (thyroid hormone T3 10 μM) on TR-β transcriptional activity in HeLa cells, measured with luciferase gene reporter assay. MTT data are means + SD of three independent experiments expressed as a percentage compared to solvent control (DMSO, 0.1% *v*/*v*). For gene reporter assay, data are reported as median + SD of three independent experiments, and the results were analyzed by Kruskal–Wallis and Dunn’s post hoc test. * represent *p* > 0.05.

**Figure 3 plants-11-03179-f003:**
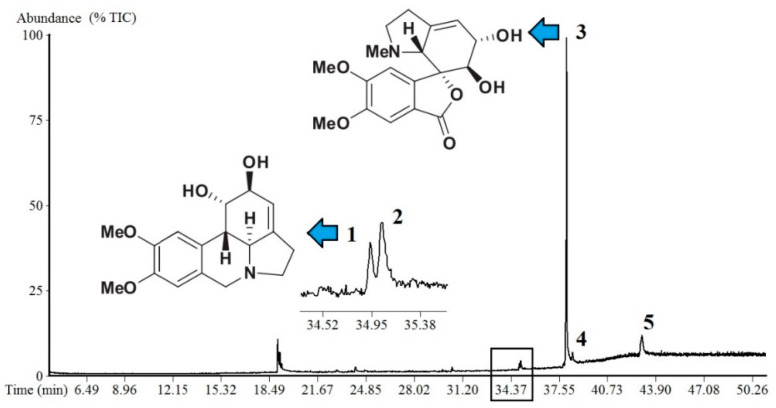
Alkaloids of *Hippeastrum stapfianum* leaves, identified by gas chromatography–mass spectrometry (GC-MS). GC-MS chromatogram with relative % of total ion current (TIC). The black box refers to peaks 1 and 2.

**Figure 4 plants-11-03179-f004:**
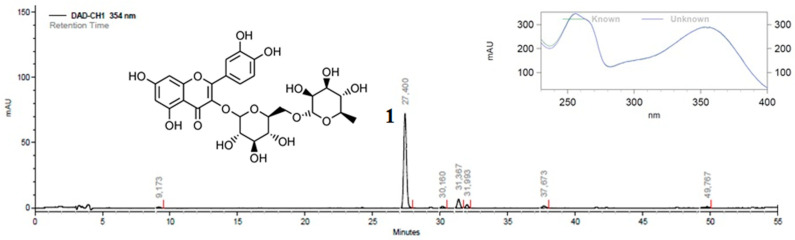
High-performance liquid chromatography (HPLC) profile of ethanol extract of *Hippeastrum stapfianum* leaves (EE) at 354 nm and UV spectra (230–400 nm) of peak number 1 and rutin overlapped.

**Figure 5 plants-11-03179-f005:**
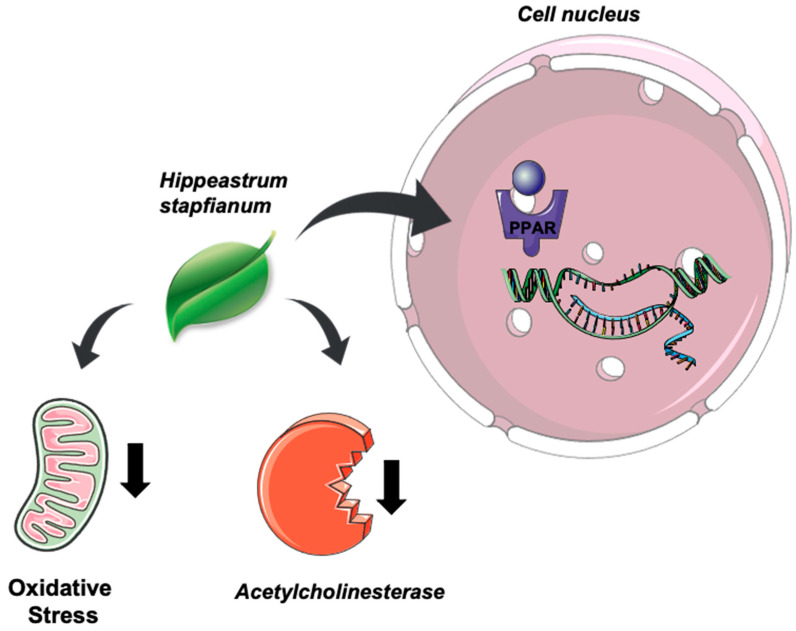
Schematic representation of *H. stapfianum* leaves modulating positivity the PPAR receptors, as well as the acetylcholinesterase downregulation and oxidative stress decrease (Servier Medical Art by Servier is licensed under a Creative Commons Attribution 3.0 Unported License).

**Table 1 plants-11-03179-t001:** Metabolites of *Hippeastrum stapfianum* identified by gas chromatography–mass spectrometry (GC-MS).

Peak	%TIC	RT	Compound	Mass Fragmentation (Relative Intensity)	Reference
1	0.8	34.9	Lycorine	226 (100), 227 (59), 250 (19), 27 (187), 286 (13), 147 (10), 228 (9), 119 (9), 248 (9)	[28]
2	0.6	35.0	Non-identified (NI-1)	343 (100), 341 (71), 344 (36), 266 (33), 40 (29), 252 (28), 196 (21), 282 (19), 310 (19), 283 (18)	-
3	94.6	38.0	7-Demethoxy-9-*O*-methylhostasine	125 (100), 96 (21), 309 (7), 311 (5), 94 (5), 126 (4), 123 (4), 124 (3), 193 (3), 82 (3)	[29]
4	1.3	38.4	Non-identified (NI-2)	311 (100), 40 (27), 310 (22), 294 (13), 309 (13), 312 (13), 296 (11), 251 (11), 208 (10), 268 (9)	[30]
5	2.7	43.0	Non-identified (NI-3)	297 (100), 254 (39), 296 (26), 252 (23), 148 (20), 298 (16), 77 (11), 295 (10), 236 (9), 196 (9)	-

All compounds were identified with mass spectra fragmentation comparison from specialized literature data. The proportion of each alkaloid is expressed as a percentage (%) of the total alkaloids measured by total ion current (TIC). These data do not express a real quantification, and NI means not identified.

## Data Availability

The original contributions presented in the study are included in the article, further inquiries can be directed to the corresponding author.

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
