# Peer review of "Hippeastrum stapfianum (Kraenzl.) R.S.Oliveira & Dutilh (Amaryllidaceae) Ethanol Extract Activity on Acetylcholinesterase and PPAR-α/γ Receptors"

_plants, 2022, doi:10.3390/plants11223179_

Round 1

Reviewer 1 Report

Figure no. 2 is not clear. Why is the HPLC analysis done on 3 different extracts, and another solvent is chosen for cell viability. Why is the chemical analysis not determined and the alcoholic extract. What standards are used?

Author Response

  1. Figure no. 2 is not clear. 2. Why is the HPLC analysis done on 3 different extracts, and another solvent is chosen for cell viability. Why is the chemical analysis not determined and the alcoholic extract. 3. What standards are used?

Thank you very much for the comments.

1)All Figures were changed for others with better quality.

2. We believe this is a misunderstanding problem. The text has been updated for better understanding. The same ethanol extract was evaluated by HPLC and used for the cell viability test.

3. For comparison, a library of flavonoids and other phenolics was used. The library was built in the same method used for the extract analysis. The major flavonoid was identified as rutin, and the identification was by retention time and UV spectra comparisons.

Reviewer 2 Report

The manuscript named “Hippeastrum stapfianum (Kraenzl.) R.S. Oliveira & Dutilh (Amaryllidaceae) ethanol extract activity in PPAR-α/γ receptors as strategic therapeutic targets for Alzheimer's disease” discusses a really interesting topic.

However, the manuscript is poor. Except many stylistical (latin words written in italics font – Amaryllidaceae or Hippeastrum stapfianum, Alzheimers Disease with little “d”) or grammatical errors (f. e. line 51 PPARs are a ligand-activated transcription factor).

Figure 2 is in a very low quality. In line, the link is given to Figure 3D, but it should be 2D. In the legend of Figure 2, the letter A is crossed, why? On the other hand, in line 86, there is another “A”.

Please, clarify the sentence in line 103 The EE result is relevant once ….. What is EE result?

Figure 3 – the abbreviation TIC without previous explaining. The explaining is firstly in Table 1.

The discussion could be written deeper.

From the results it is readable that the extract has an impact on HeLa cells and that it modulates PPARα and PPARγ receptors. However, there is no proof that it has some anti-Alzheimer activity, thus, the title and the statements should be more careful and not so strong.

Author Response

1. Except many stylistical (latin words written in italics font – Amaryllidaceae or Hippeastrum stapfianum, Alzheimers Disease with little “d”) or grammatical errors (f. e. line 51 PPARs are a ligand-activated transcription factor).

Thank you very much for the comment. The stylistic and grammatical errors have been corrected, and the “d” in Alzheimer's disease should remain lowercase. We checked on all mentioned occasions. Concerning the botanical terms, we followed International Plant Name Index, where species names are in italics but not the authors (IPNI, https://www.ipni.org/n/77203201-1).

All the text was revised by a native English speaker (Dr Fagg, one of the authors)

2. Figure 2 is in a very low quality. In line, the link is given to Figure 3D, but it should be 2D. In the legend of Figure 2, the letter A is crossed, why? On the other hand, in line 86, there is another “A”.

Thank you for your comment. All Figures were changed for others with better quality. In line, the link Figure 2D was corrected. In the legend of Figure 2, the crossed letter A was a typing mistake and was corrected. The exceeding A was also removed.

3. Please, clarify the sentence in line 103 The EE result is relevant once ….. What is EE result?

 The text was changed to “The IC50 result for EE is relevant once an extract was used and not an isolated compound”.

4. Figure 3 – the abbreviation TIC without previous explaining. The explaining is firstly in Table

 The explaining was added in the Table caption and the text.

5. The discussion could be written deeper.

Thank you for your comment. The discussion was improved. The revised manuscript is with MS change control, so all changes and text insertions can be easily observed

6. From the results it is readable that the extract has an impact on HeLa cells and that it modulates PPARα and PPARγ receptors. However, there is no proof that it has some anti-Alzheimer activity, thus, the title and the statements should be more careful and not so strong.

We completely agree with the comment. We adapted the title according to the results obtained in the present study: “Hippeastrum stapfianum (Kraenzl.) R.S.Oliveira & Dutilh (Amaryllidaceae) ethanol extract activity in the acetylcholinesterase and PPAR-α/γ receptors”.

Reviewer 3 Report

Evaluation of the manuscript by Kicia Karinne Pereira Gomes-Copeland entitled “Hippeastrum stapfianum (Kraenzl.) R.S.Oliveira & Dutilh (Amaryllidaceae) ethanol extract activity in PPAR-α/γ receptors as strategic therapeutic targets for Alzheimer's disease” sent to Plants.

The manuscript contains valuable interesting data and after minor improvements may be published in Plants.

Line 17: what do you mean by biological potential?

Line 18: molecule candidates for Alzheimer's Disease, a multifactorial pathology. Please, redraft. It is vague.

Line 20: the action on nuclear receptors PPAR-α and PPAR-γ. Please be more exact in such statements. What action?

Why do you concentrate on Alzheimer disease? acetylcholinesterase has got many other functions. E.g. acetylcholinesterase works by hydrolyzing the ester bond of acetylcholine to produce acetic acid and choline. This process regulates the presence of acetylcholine after it has been released from the nerve cell into the neuromuscular junction to prevent excessive muscle contraction. But generally is it OK. What other, apart from acetylcholinesterase activity, parameters could be specific for prevention of AD?

Figure 2 and others: could you bring pictures of better quality? If possible.

Lines 145-157: those sentences should be put rather into introduction, if any. In the discussion focus on mechanisms you analysed in in vitro assays.

Please expand the discussion. Now, it is very short.

Author Response

1. Line 17: what do you mean by biological potential?

Thank you for your comment. The expression "Biological potential" was removed and the Abstract text was changed.

2. Line 18: molecule candidates for Alzheimer's Disease, a multifactorial pathology. Please, redraft. It is vague.

 Thank you for your comment. The phrase was removed and the Abstract was changed as below:

"Hippeastrum stapfianum (Kraenzl.) R.S.Oliveira & Dutilh (Amaryllidaceae) is an endemic plant species from the Brazilian Savannah with biological and pharmacological potential. This study evaluated the effects of ethanol extract from H. stapfianum leaves on acetylcholinesterase enzyme activity and the action on nuclear receptors PPAR-α and PPAR-γ. A gene reporter assay was performed to assess the PPAR agonist or antagonist activity with a no-toxic dose of H. stapfianum ethanol extract, the antioxidant capacity by DPPH scavenging, and fosfomolybdenium reduction assays, and the identification of H. stapfianum chemical composition by gas chromatography-mass spectrometry (GC-MS) and HPLC. The ethanol extract of H. stapfianum activated PPAR-α and PPAR-γ selectively, inhibited the acetylcholinesterase enzyme, and presented antioxidant activity by in vitro assay. The major compounds identified were lycorine, 7-demethoxy-9-O-methylhostasine, and rutin. Therefore, H. stapfianum is a potential source of drugs for Alzheimer's Disease due to its ability to activate PPAR receptors, acetylcholinesterase inhibition, and antioxidant attributes."

 3. Line 20: the action on nuclear receptors PPAR-α and PPAR-γ. Please be more exact in such statements. What action?

       "Considering this important comment by the revisor, we reformulated the text to “This study evaluated the effects of ethanol extract from H. stapfianum leaves on acetylcholinesterase enzyme activity and the action on nuclear receptors PPAR-α and PPAR-γ. A gene reporter assay was performed to assess the PPAR agonist or antagonist activity with a no-toxic dose..."

4. Why do you concentrate on Alzheimer disease? acetylcholinesterase has got many other functions. E.g. acetylcholinesterase works by hydrolyzing the ester bond of acetylcholine to produce acetic acid and choline. This process regulates the presence of acetylcholine after it has been released from the nerve cell into the neuromuscular junction to prevent excessive muscle contraction. But generally is it OK. What other, apart from acetylcholinesterase activity, parameters could be specific for prevention of AD?

Thank you very much for your comment. In fact, there is a rationale that may have been forgotten in the introduction, about why the research is based on Alzheimer's disease. Although Amaryllidaceae alkaloids present several pharmacological activities, they have been investigated as Alzheimer's disease potential drugs for decades. Galantamine is even used clinically with benefits in cognition/memory in these patients. This information was highlighted to justify the selected theme better.

5.Figure 2 and others: could you bring pictures of better quality? If possible.

All Figures were changed for others with better quality

 6. Lines 145-157: those sentences should be put rather into introduction, if any. In the discussion focus on mechanisms you analysed in in vitro assays.

 We consider this observation relevant and thus adjusted the introduction and discussion as suggested. The revised manuscript is with MS changes control, so all revisions can be easily accessed.

 7.Please expand the discussion. Now, it is very short.

The discussion was improved and expanded.

Reviewer 4 Report

Lines 48-50: this sentence should be appropriately referenced. The same for lines 58-59

Sections 4.6 – 4.8: please state the reagent sources for these experiments.

No statistical analysis sub-section is available in section 4. Methods and software should be shortly described (Kruskal–Wallis and post-hoc tests are reported in section 2 for the MTT, they should be mentioned; why was not ANOVA used and how its inappropriateness was concluded; how were IC50 values estimated).

Section 2 reports HPLC results, whereas section 4 contains no info on the HPLC method used.

Lines 120-121: “In summary, the EE of H. stapfianum positively modulates the PPAR receptors, inhibits the acetylcholinesterase activity, and  shows  interesting  antioxidant  capacity.” Actually, as shown by results in section 2.1 (in particular Figure 2), the effect for both PPAR receptors are rather minimal, and this aspect should be critically discussed. One should not expect an effect similar in size with that of rosiglitazone or bezafibrate.

The same holds true for lines 173-175: “In the present study, we  demonstrate  the  effects  of  EE  on  acetylcholinesterase  inhibition,  antioxidant  capacity,  and PPAR-α and PPAR-γ activation, which occurs selectively”. A scientific report should not strive to report positive findings, but rather to report true findings. Therefore, a more objective description would along the lines of “In the present study, we evaluated the effects…”, then reporting objectively that the effects on the PPAR receptors are quite minor (and unlikely to be of clinical significance). The same for lines 197-199: “Taken together, the present study suggests the chemical compound of the EE as a potential PPAR  activity, contributing to acetylcholinesterase and oxidative stress modulation”. This statement is little based in the results reported, and it should rephrased to correctly reflect the findings. The same holds true for lines 206-207.

The authors also almost gloss over the fact that the IC50 for the DPPH was about 10 times higher than that of ascorbic acid (AA), which also shows that although there is antioxidant effect, it is considerably lower to that of AA.  

Author Response

1. Lines 48-50: this sentence should be appropriately referenced. The same for lines 58-59

We appreciate the valuable suggestion. Additional references were added for a better understanding and the text was improved.

 2.Sections 4.6 – 4.8: please state the reagent sources for these experiments.

 The reagent sources were added.

 3. No statistical analysis sub-section is available in section 4. Methods and software should be shortly described (Kruskal–Wallis and post-hoc tests are reported in section 2 for the MTT, they should be mentioned; why was not ANOVA used and how its inappropriateness was concluded; how were IC50 values estimated).

A topic regarding statistical analysis was added in section 4 (Material and Methods) to explain all these questions, as presented below.

"4.11 Statistical and data analysis

“The viability of the cells was assessed by mitochondrial activity presented by percentage related to DMEM media (control). The gene reporter Luciferase assay results were presented in a dot plot with standard deviation (SD). Considering that some groups presented non-normally distributed data, statistical differences among groups were tested by Kruskal–Wallis and post hoc Dunn’s test (GraphPad Prism Software, version 5.01). The activation rate of the transcription of the groups treated with the positive controls or the extract was compared to the group treated with vehicle (control). The significance criterion for all analyses was p < 0.05. The results of the in vitro acetylcholinesterase inhibition and antioxidant capacity assays were tested by one-way analysis of variance (ANOVA) followed by Dunnett’s post hoc test, and p values < 0.05 were considered significant.”

4. Section 2 reports HPLC results, whereas section 4 contains no info on the HPLC method used.

Thank you very much for your comment. The Material and Methods section was fully revised and the information was added in the appropriate section, with the respective reference (see below)

 “4.10. HPLC-DAD-UV analysis

EE was solubilized in methanol (1 mg/mL), and 10 μL of this sample was analyzed using LaChrom Elite HPLC system (Hitachi®, Tokyo, Japan) liquid chromatograph equipped with L2130 pump, auto-sampler L2200, L2300 column oven was set at 25 °C and a L2455 diodo array detector (DAD) (Hitachi®, Tokyo, Japan). The C-18 column (5 μm, 150 mm × 4.6 mm) was used in combination with an appropriate guard column (4.0 mm × 4.0 mm; 5 μm of particle size) (Merck®, Germany). The analysis was performed at a wavelength fixed at 354 nm. The eluents used were aqueous phosphoric acid (1%) (solvent A) and acetonitrile (solvent B). The gradient employed was 90% A and 10% B for 0 min, 70% A and 30% B for 40 min, 50% A and 50% B for 50 min, 90% A and 10% B for 51 min, and 90% A and 10% B for 55 min at a flow rate of 0.6 mL/min. Data acquisition was performed using ExChrom Elite software (version 3.3.2 SP1) (Scientific Software Inc.). The compounds present in the extract were compared according to their UV–Vis spectra (similarity index > 0.99) and retention times with commercial standards.”

5.Lines 120-121: “In summary, the EE of H. stapfianum positively modulates the PPAR receptors, inhibits the acetylcholinesterase activity, and  shows  interesting  antioxidant  capacity.” Actually, as shown by results in section 2.1 (in particular Figure 2), the effect for both PPAR receptors are rather minimal, and this aspect should be critically discussed. One should not expect an effect similar in size with that of rosiglitazone or bezafibrate.

 Thank you for your comment. We agree with the observation, and we adjusted the topic in the results:

 "The dose-response analysis suggested that the EE markedly statistically improved the transcriptional activity of PPAR-α and PPAR-γ at 250 and 300 μg/mL when compared to vehicle  (Figures 2 B and C). However, compared with the positive control (rosiglitazone for PPAR-γ or bezafibrate for PPAR-α), the agonist activity was slight, suggesting modulation partial in response to these receptors."

 6.The same holds true for lines 173-175: “In the present study, we  demonstrate  the  effects  of  EE  on  acetylcholinesterase  inhibition,  antioxidant  capacity,  and PPAR-α and PPAR-γ activation, which occurs selectively”. A scientific report should not strive to report positive findings, but rather to report true findings. Therefore, a more objective description would along the lines of “In the present study, we evaluated the effects…”, then reporting objectively that the effects on the PPAR receptors are quite minor (and unlikely to be of clinical significance). The same for lines 197-199: “Taken together, the present study suggests the chemical compound of the EE as a potential PPAR  activity, contributing to acetylcholinesterase and oxidative stress modulation”. This statement is little based in the results reported, and it should rephrased to correctly reflect the findings. The same holds true for lines 206-207.

 We completely agree with the comment. We revised all these topics:           

The data obtained here suggest that compounds from EE present a potential PPAR activity. In addition, the results suggested acetylcholinesterase and oxidative stress modulation promoted by Hippeastrum stapfianum extract."

“Hippeastrum stapfianum ethanol extract (EE) can potentially be a new therapeutic option for Alzheimer's disease treatment due to the ability to activate PPAR receptors selectively. In addition, EE inhibited the AChE and presented antioxidant capacity under the conditions evaluated.”

 The revised manuscript is with MS change control, so, all the changes can be easily accessed.

7.The authors also almost gloss over the fact that the IC50 for the DPPH was about 10 times higher than that of ascorbic acid (AA), which also shows that although there is antioxidant effect, it is considerably lower to that of AA. 

To better understand, we change the text to:

 “In the present study, we highlight another finding: the antioxidant potential evaluated by two in vitro models, DPPH and phosphomolybdenum. EE revealed an EC50 of 61.88 ±0.43 µg/mL. Although this result seems irrelevant compared to ascorbic acid, it is important to point out that EE refers to a complex mixture of several diverse compounds being compared to a pure substance.”

Round 2

Reviewer 1 Report

The requested changes have been made

Author Response

Dear Reviewer,

Thank you for expend your time helping us to improve the manuscript. 

Reviewer 4 Report

The text has definitely been improved. In my view, though, the text still tends to over-emphasize the effect on the PPAR receptors, although its size is very limited. Otherwise, it can be published.

Author Response

Dear Reviewer,

We agree with you about the emphasis on nuclear receptors. In fact, this investigation is a novelty in the field of chemical-biological studies of extracts from Amaryllidaceae. In this sense, providing the majority of attention to these investigated targets was intentional. Small changes in the text were done to reduce such emphasis.